# Consumers’ Perception towards Organic Products before and after the COVID-19 Pandemic: A Case Study in Bihor County, Romania

**DOI:** 10.3390/ijerph191912712

**Published:** 2022-10-05

**Authors:** Anca Monica Brata, Aurelia Ioana Chereji, Vlad Dumitru Brata, Anamaria Aurelia Morna, Olivia Paula Tirpe, Anca Popa, Felix H. Arion, Loredana Ioana Banszki, Ioan Chereji, Dorin Popa, Iulia C. Muresan

**Affiliations:** 1Department of Engineering of Food Products, Faculty of Environmental Protection, University of Oradea, 26 Gen. Magheru St., 410087 Oradea, Romania; 2Department of Animal Husbandry and Agritourism, Faculty of Environmental Protection, University of Oradea, 26 Gen. Magheru St., 410087 Oradea, Romania; 3Faculty of Medicine, “Iuliu Hatieganu” University of Medicine and Pharmacy, 400000 Cluj-Napoca, Romania; 4Department of Economic Sciences, University of Agricultural Sciences and Veterinary Medicine Cluj-Napoca, 3-5 Manastur Street, 400372 Cluj-Napoca, Romania; 5Faculty of Medicine and Pharmacy, University of Oradea, 410087 Oradea, Romania

**Keywords:** organic products, consumer behavior, COVID-19

## Abstract

Organic products have grown ever more popular in recent years due to increased concerns regarding one’s health, the environment, and sustainability. The COVID-19 pandemic has only accelerated this growth, prompting producers to adapt to a rapidly growing market while maintaining the same standard of quality. We have conducted a questionnaire-based pilot study on 190 organic food consumers from Bihor county, Romania, in order to analyze the main factors influencing customers’ beliefs regarding organic food consumption, as well as assess the extent to which their consumption frequency changed after the pandemic. A principal component analysis was performed, dividing the factors into two categories referring to intrinsic and extrinsic characteristics of the products, respectively. It was noticed that the most important cues were ranked the same by our 190 respondents, regardless of their socio-demographical background. At the same time, people who consumed organic products more frequently before the pandemic have either maintained or increased their consumption, while more indifferent consumers maintained or decreased the proportion of organic foods in their diet.

## 1. Introduction

During the past decades, many changes have interfered with social and economic development, most of them regarding environmentalism, health, and quality of life, a phenomenon also depicted by the mobilization of consumers through organic values [1]. As a result, the link between socioeconomic development and the presence of a values-driven mentality toward the environment, food quality, and health has never been more evident [2].

Additionally, social movements and dynamics, such as the COVID-19 pandemic and ethical consumption, are capable of influencing the development of certain values and dietary habits, changing the consumers’ perceptions and attitudes regarding food consumption [1,2]. Existing problems regarding sustainable consumption and food waste are growing more evident in today’s society, with organic food consumption as a possible way to address these issues [3,4].

According to the Medical Dictionary, organic food represents a category of food that, in the purest form, is grown without chemical fertilizers, pesticides, genetically modified organisms (GMOs), or other potentially harmful substances, being sold to the consumer without adding preservatives and synthetic food enhancers [5,6]. As a result, it decreases the production of carbon dioxide (CO2) and other polluting substances while also offering consumers food with a high nutritional value [5]. It has become more and more popular in recent years, especially during the pandemic, when people began taking a growing interest in fresh, locally produced, and healthier products [7,8]. Additionally, since consumer market behavior has changed as a result of personal ideas and opinions about food quality and safety merging with expanding environmental and pollution concerns, the organic food business now has a major window of opportunity for growth [4,9,10,11].

The consumption of organic products has also been proven as relevant for public health, being recognized as an important tool in controlling and preventing various diseases which have seen a significant rise in incidence over the years, such as obesity and type 2 diabetes mellitus [12,13]. Nutritional treatment represents the first line of therapy for these diseases, further underlining the importance of a balanced diet with healthy products [12,13]. 

Growing health concerns during the COVID-19 pandemic have raised consumer awareness of lifestyle and dietary choices, with organic products becoming more and more popular [14]. Concerns about the environment and sustainability have also grown in recent years, intensifying during the pandemic and fueling a spike in interest in organic products [15,16,17,18].

Thus, in the light of increased awareness regarding food consumption and one’s health, the aim of our study was to assess the factors influencing consumers’ behavior regarding organic product purchasing in correlation with the socio-demographical characteristics of our respondents, as well as to determine whether the COVID-19 pandemic had an influence on the frequency of consumption of these types of foods. 

To this extent, the findings of this study could inform public health and environmental experts regarding the perception of consumers towards organic products, as well as how this has changed during the COVID-19 pandemic. Moreover, this research could also prove useful to organic food producers in order to understand the consumers’ needs and adapt their production accordingly. This research could also be relevant for the main stakeholders in this domain, taking increased interest in growing the production and consumption of organic products, leading to economic, social, and environmental benefits for the local community. Although the organically cultivated surfaces have recorded a steady growth during the past years in Romania, reaching 580.819,13 ha in 2021 from 182.706 in 2010 [19], organic food retail sales are still lower than in other European countries [20], with most of the organic food produced in Romania being exported [21]. Thus, there is a significant need to identify factors influencing organic food consumption in order to build sustainable strategies and support organic producers. 

## 2. Consumers’ Perception Regarding Organic Food Products—A Literature Review

Existing research has identified a series of factors influencing consumers’ behavior regarding food consumption and has further classified them into intrinsic and extrinsic characteristics of the products [22,23,24,25,26]. To this extent, in the first category have been included factors related to the product itself, consisting of features of the food (whether it is healthy, taste, quality, nutritional value, whether it is fresh or has a long shelf-life), and these characteristics cannot be changed without altering the product [12,13,14,15,16,17,18,19,20,21,22,23,24,25,27]. On the other hand, the extrinsic cues are related to the market (income, price), social beliefs and norms (fashion, tradition, behavior of other consumers), environmental motives (environmental protection, animal welfare), availability and location of purchase [12,13,14,15,16,17,18,19,20,21,22,23,24,25,28,29]. Research has also concluded that the lack of pesticides [30,31], as well as GMOs [32], represent a strong motive among consumers in order to choose an organic product instead of a conventional one. When it comes to the decision to purchase an organic product, studies have concluded that the intrinsic characteristics of the products influence this process regardless of the socio-demographical characteristics of the consumers [33,34,35,36,37,38,39,40,41]. Factors such as health benefits, superior quality, as well as sensory characteristics have ranked highest on consumers’ priority lists worldwide [33,34,35,36,37,38,39,40,41]. Additionally, more educated customers and higher-income individuals tend to consume organic products also due to environmental beliefs, as well as a reduced perception of the price of these products as a barrier [33,34,35,36,37,38,39,40,41].

Moreover, studies have underlined the importance of the location of production, with research confirming that people who take a greater interest in this matter and purchase more from local producers also tend to consume more organic products [23,25]. This comes from the consumers’ belief that local organic products are of greater quality while also contributing to local economic development [23,25].

Nevertheless, several differences in perception regarding organic food consumption have also been recorded, but the high quality and health advantages of these kinds of products continue to be widely recognized [42,43,44]. Thus, Azzurra et al. concluded that concerns regarding food sustainability, as well as increased interest in a sustainable lifestyle, determined a greater consumption of organic products [42]. Moreover, Lang and Rodriguez analyzed the perception of consumers and factors influencing their decision-making process regarding both organic certified and non-certified foods [44]. The study concluded that customers are inclined to pay a higher price for organic products if certain requirements are met, such as lack of GMOs and hormones, sustainable agricultural influences, and certification [44]. However, a higher price could also represent a barrier to organic food consumption [45]. A study conducted by Bryła on Polish respondents confirmed the previous findings regarding the quality requirements for organic products (healthiness, superior taste, quality, and food safety considerations), while the higher price compared to other food products represented the main barrier for Polish consumers [45]. Additionally, women were more likely to prefer organic food, as well as respondents with higher education levels and higher-income individuals [45]. These findings were confirmed by a study conducted in Korean households, concluding that the vast majority of Korean women had a significant awareness regarding organic food (94.1%), with 71.7% of them purchasing it regularly [46]. Despite the popularity of organic products in Korean households, the main barrier to purchasing such products was the price, with 95.9% of the respondents considering them too expensive [46].

A large German study assessed the main ways consumers resort to when it comes to informing themselves regarding the nutritional value of the food they purchase [47]. Thus, the majority of the respondents obtained their nutritional information from radio and television, as well as from professionals in the field, such as doctors and pharmacists [47]. Nevertheless, a significant proportion of the respondents took into account the recommendations of their colleagues, friends, and other family members when purchasing food [47].

Research also revealed that younger consumers tend to be more interested in the environment and the consequences their actions have on it and have a generally positive attitude towards green living [48,49,50]. Additionally, this further influences their purchase decisions regarding green and organic products, with studies directly correlating environmental concerns and attitudes with the purchase intention of such products [50,51]. Moreover, they are more likely to pay additional attention to the product’s label [48]. Customers might educate themselves on the different food attributes, nutritional value, and production process to some extent through labeling [48,52,53]. With a substantial number of research indicating differences in perception and purchase frequency regarding organic products [48,52,53,54,55], the education level of consumers is particularly important. Thus, in general, consumers with lower educational levels tend to consume less organic food and have a decreased interest in sustainability and the environment, while people with a college degree are perceived to be the most consistent consumers of organic products [48,52,53,54,55].

Consumers’ behavior towards healthy food and organic products, in particular, could also be influenced by certain advertisement campaigns, which would increase awareness regarding the impact food consumption has on the environment. To this extent, Mo et al. analyzed the impact of “green ads” campaigns on respondents with different levels of environmental concerns. The study concluded that the more interested a consumer was in environmental issues, the higher the perceived effectiveness of the ad [56]. 

When it comes to the main types of organic food preferred by consumers, studies have revealed that fresh fruits and vegetables rank highest in customers’ preferences, as well as bread, cereal, and dairy products [57,58,59,60], even though there are differences between countries, with fruits and vegetables being preferred in Serbia and Spain [58,59,60], while Croatians consumed more organic bread and cereal products [57]. Danish consumers preferred organic milk and dairy products over organic fruits, vegetables, bread, cereals, and meat [58].

Health concerns during the COVID-19 pandemic have only increased, with more and more people taking an increased interest in their lifestyle and their impact on the environment [43,61,62,63]. In order to better understand this phenomenon, it is crucial to analyze the extent to which the pandemic has influenced the decision-making process regarding food consumption, especially organic products. With the consumption of organic products recording a significant increase during the past years [17], the increased awareness regarding one’s health and diet due to the pandemic has only accelerated this trend [15,64].

De Barcker et al. conducted a study on participants from 38 countries regarding the changes occurring in planning, selecting, and preparing healthy foods during the initial lockdown in 2020 [65]. Thus, it was reported that financial stress due to the pandemic was associated with a decrease in preparing and planning healthy foods, with less money available for shopping, causing a negative change in both men and women. Moreover, the closure of restaurants and bars caused an increase in healthy meal planning and preparation for women but a decrease for men. Women have reported a significant increase in preparing healthier foods due to having more time and working from home during the lockdown compared with men. With schools being closed and public gatherings restricted as well, both men and women reported an increase in selecting healthier foods [65]. 

To this extent, a study conducted by Xie et al. on Chinese consumers shortly after the outbreak of the COVID-19 disease has underlined the respondents’ reactions regarding organic food consumption, as well as game meat [61]. Thus, due to health concerns, organic products have become more appealing to Chinese consumers, while the consumption of game meat has decreased significantly [61]. Another study conducted on Chinese respondents roughly two years into the pandemic has reported that the popularity of organic foods in China has maintained steady growth, with these types of products being purchased regularly by Chinese customers [43]. Moreover, Jiao et al. analyzed the impact of the first lockdown on the dietary habits of Chinese consumers, concluding that, in general, the diet of the Chinese respondents remained unchanged, with a decrease in the consumption of salty foods and alcoholic beverages [66]. The study also showed that the higher the socioeconomic status of the consumers, the more interested they were in maintaining a healthy diet [66]. Wachyuni and Wiweka have also observed an increased interest of Indonesian consumers in organic products and cooking at home due to the concerns and restrictions imposed during the pandemic [62].

Additionally, Busch et al. conducted a study on German respondents regarding the influence of the pandemic on their shopping decision, concluding that the stable shelf life of the products has become more important for the consumers, together with the health benefits and regional origin [67]. Moreover, more than 80% of the respondents purchased more organic food than before the pandemic [67].

Nevertheless, there are some studies that have concluded that the diet of consumers has not significantly changed during the pandemic [68,69]. Thus, Hansmann et al. reported that 77.3% of their Swiss respondents had the same consumption patterns, with less than a quarter (22.3%) changing their diets [69]. Notably, no significant differences between genders were recorded; however, younger people were more likely to have changed their food consumption patterns, compared with the older generation, possibly due to the more stable habits of the older generations [69]. Moreover, the majority of the ones who changed their diets claimed they purchased significantly more organic products (56.2%) [69].

Although some changes have occurred regarding consumers’ preferences for the different varieties of organic products, such as an increase in demand for cereals, flour, and pasta, fruits and vegetables remained the most popular organic food products during the pandemic [40,55]. Based on those mentioned above, the following research questions have been elaborated: What are consumers’ perceptions of organic products? What is the relationship between the socio-demographical characteristics of the respondents and the identified components in the analysis? To what extent has the COVID-19 pandemic influenced the frequency of consumption of organic products, and what categories recorded the most significant changes?

## 3. Materials and Methods

The main objective of the current study was to assess the consumers’ perception of organic products, as well as identify the main factors influencing the consumption of organic products. Additionally, we investigated the impact of the COVID-19 pandemic on the frequency of consumption of this type of product.

### 3.1. Research Methodology and Questionnaire Design

To achieve the aim of the research, an online survey was conducted in May–July 2022 among residents from Bihor County of Romania. A total number of 225 questionnaires were collected, from which 190 were validated for the current research since the respondents declared that they were consuming organic products. The research instrument consisted of 3 main sections: (i) socio-demographic characteristics; (ii) factors affecting organic food products consumption; (iii) consumers’ behavior towards organic food products before and after the COVID-19 pandemic. To identify the factors that affect the consumption of organic food products, a set of 19 items adapted for previous research [70,71,72] was used. The 19 items were related to features of the food products (whether they are healthy, their taste, quality, and nutritional value, whether they are fresh or have a long shelf-life, whether they are without GMOs and without additives, and whether they had superior quality) on the 1 hand, and to the market, social believes, availability, on the other hand. Each of the 19 items was evaluated on a scale from 1 to 5, where 1 means not all important, and 5 means very important. A study of 10 consumers was conducted in order to test the feasibility of the research instrument. Based on the pilot study, the research instrument was reviewed, and the items related to food consumption and categories of organic products were updated. 

### 3.2. Sample Size, Data Collection, and Organic Food Industry in Bihor County

A non-probability convenience sample of 190 consumers of organic products from Bihor county, Romania, was established. The sample size met the criteria of 5:1 subjects to item [73]. Bihor county belongs to the Northwestern region of Romania, with an area of 7844 km^2^, out of which 309.265 hectares of arable land [74]. There has been a significant increase in organic-certified agricultural areas in the last years, from 1.931 hectares in 2010 to 2.462 hectares in 2021, with 35.3% of these areas being destined for human consumption [75,76]. Furthermore, Bihor county was chosen as the focus of the research because, even though it has the largest area designed for agriculture in the Northwestern region of Romania, smaller neighboring counties have larger areas destined for organic agriculture [75,76]. Additionally, in 2020, there were only 292 organic food operators in the county, well below the national average [74,75].

The data was collected using a self-administrated online survey, during which the participants were informed about the aim of the research and gave their consent regarding the processing of their personal data in accordance with the General Data Protection Regulation of the European Union. Ethical review and approval were waived for this study due to the fact that participation was voluntary, and all data were anonymous. When it comes to informed consent, this was obtained from all respondents involved in the study. Using a questionnaire-based method, all respondents had to provide their consent in order to proceed to the actual set of questions.

### 3.3. Data Analysis

Data were analyzed using SPSS 26.0 software package (SPSS Inc., Chicago, IL, USA). The socio-demographic profile of the respondents, as well as the frequency of consumption, were assessed using descriptive statistics. When it comes to the dimensionality of the 19 items, principle component analysis (PCA) was performed in order to evaluate the influence each item had on organic product consumption. The two retained factors had an eigenvalue over one, with Cronbach’s alpha coefficient of 0.964, indicating a good internal consistency of the items. Barlett’s test of sphericity was significant (Chi-Square = 3831.668; *p* < 0.001), and the Kaiser-Meyer-Olkin measure of sampling adequacy retained a value of 0.947, confirming that the considered data were suitable for PCA. The varimax rotation of the 19 variables resulted in a 2-component solution explaining 73.323% of the total variance, with factors with an eigenvalue greater than 1 being selected.

Moreover, the Mann-Whitney U and the Kruskal-Wallis tests were performed in order to assess the relationship between the socio-demographic characteristics of the respondents and the main factors resulting from the PCA.

## 4. Results

### 4.1. The Socio-Demographical Profile of the Respondents

Out of the total number of respondents, the vast majority were female (71.1%), in comparison with male (28.9%), and were residing in urban areas (64.2%), while 35.7% lived in rural areas. When it comes to education level, more than half of the respondents had a university degree (38.4%) or postgraduate degree (36.3%). Additionally, the proportion of participants belonging to the 26–35 (3.7%), 36–46 (23.2%), and 46–55 years age groups (23.7%) was fairly distributed. Regarding the income levels of the respondents, most of them reported household incomes of more than 3.000 RON monthly, with 36.3% reporting more than 5.000 RON per month, while only 7.4% of the households in question reported less than 2.000 RON monthly. The socio-demographic characteristics of the respondents are illustrated in Table 1.

### 4.2. Principal Component Analysis

The two factors resulting from the PCA have been detailed in Table 2, together with their corresponding items. The first component has been labeled “Intrinsic cues” and consists of factors depending on the biological products. The factor consisted of 10 variables and explained 68.120% of the total variance and had a reliability coefficient of 0.895, with a mean of 4.07 ± 1.098. Thus, the consumers took most into account whether the organic products had a superior quality compared to other products (4.23 ± 1.211), were fresh (4.22 ± 1.187), healthy (4.15 ± 1.238), or natural (4.07 ± 1.237), while also considering their shelf life (4.17 ± 1.197). Out of the variables belonging to this component, the nutritional value of the product was ranked as the least important according to our respondents (3.74 ± 1.245).

The second component of our PCA analysis consisted of extrinsic characteristics of the organic products, and accounted for 11.143% of the total variance, and had a reliability coefficient of 0.922, with a mean of 3.32 and SD of 1.048. Out of the nine analyzed items, consumers paid the most attention to the price of the organic products (3.65 ± 1.304), their variety (3.52 ± 1.263), country of origin (3.51 ± 1.352), and whether they were available in a supermarket or not (3.51 ± 1.383). Moreover, the producer’s logo (3.18 ± 1.350), the brand of the product (3.06 ± 1.302), and the packaging (2.95 ± 1.344) were considered the least important aspects that determined the purchase of an organic product.

### 4.3. Relationship between the Socio-Demographical Characteristics and the PCA Results

Furthermore, the Mann–Whitney-U and the Kruskal-Wallis tests were performed in order to identify and assess any significant differences between the socio-demographical characteristics of the respondents and the evaluated factors resulting from the PCA. Thus, we presented the detailed analysis and relationship between these characteristics and the two components from the PCA in Table 3. 

When it comes to factors influencing the decision to purchase a certain type of organic product, it is notable that our study did not find any significant differences between the intrinsic characteristics of the said product and the socio-demographical features of our respondents. It is apparent that, regardless of their social background, consumers tend to pay attention to and focus on the same characteristics of the product when choosing it. Thus, the only significant differences were recorded in relationship with the extrinsic characteristics of the organic products, in particular the education level (*p* = 0.031; *p* < 0.05) and the monthly household income (*p* = 0.011; *p* < 0.05). To this extent, the respondents who had a university degree were more likely to rely on extrinsic cues of the products (3.56 ± 1.043), followed by those who graduated high school (3.27 ± 1.084), while these characteristics were least influential on consumers with a postgraduate degree (3.1 ± 1.020). 

Additionally, consumers with the highest income levels were least likely to rely on the extrinsic features of the product when deciding to purchase an organic product (3.08 ± 1.002). In comparison, respondents who earned between 2.001 and 3.000 RON monthly were the ones who took into account the most of these types of features (3.75 ± 1.112).

### 4.4. The Influence of the COVID-19 Pandemic on the Frequency of Consumption of Organic Products

Furthermore, we investigated the effect of the COVID-19 pandemic on the frequency of consumption of organic products among our respondents. These results showing how the pattern changed are detailed in Table 4 and Table 5.

Overall, out of the total number of respondents, the majority reported no significant differences regarding the consumption frequency of organic food products before and after the pandemic (60.5%). Moreover, the percentage of consumers who affirmed an increased (21.1%) or a decreased interest (18.4%) in these types of products was rather similar.

Performing a more detailed analysis, most of the respondents reported buying these types of products at least once a week (16.3%), with the majority consuming them more times a week (47.9%), while some preferred them daily (13.7%). It is notable that only a small percentage of the respondents consumed organic food products once every 6 months (3.7%) or even more seldom (1.6%).

Additionally, most of the investigated consumers reported no changes in terms of frequency of buying organic products during the pandemic. This is more evident among the respondents who reported consuming these products daily (69.2%), more times a week (71.4%), or weekly (45.2%). The small percentage of respondents (3.8% and 7.7%, respectively) who reported a decrease in the frequency of consumption among the two categories who consumed organic food products most frequently is also notable. 

In comparison, people who did not have a habit of consuming these types of products (once every 6 months or even more rarely than that) reported a significant decrease in terms of frequency- 57.1% and 33.3%, respectively. 46.9% of the respondents who belonged to the “2–3 times a month” category reported decreased consumption of organic products after the pandemic, slightly more than those who revealed no significant changes in their pattern of consumption (43.8%), while only 9.4% started consuming organic products more frequently. A more detailed analysis regarding how the frequency of consumption of the main types of organic products investigated is presented in Table 6.

Thus, it can be observed that the frequency of consumption remained the same among all types of organic products preferred by the respondents. It is noteworthy that some classes have recorded a more significant decrease in consumption than an increase, such as meat (24% vs. 22.4%), canned vegetables and fruits (31.4% vs. 22.5%), cereals (23.9% vs. 20.1%), and sweets (27.7% vs. 19.3%). Additionally, more consumers have increased the purchase of several types of organic products rather than reducing the frequency of consumption, in the cases of dairy (25.2% vs. 19.6%), fresh vegetables (20.6% vs. 18.8%), and fruits (20.1% vs. 19.6%).

## 5. Discussion

Increased interest regarding food safety, balanced nutrition, and sustainability has become ever clearer during the past years, with the COVID-19 pandemic accelerating this growth [7,8]. Consumers began to inform themselves more regarding the types of products they purchase, thus leading to an increase in organic food consumption [78,79,80] due to the well-established superior quality, nutritional value, and being more environmentally friendly compared to other products [5,6].

When it comes to the main attributes consumers pay attention to when purchasing organic products, the current research confirmed previous findings in this domain, with the intrinsic characteristics of the product being the most important ones when deciding to buy a certain organic product, such as superior quality, being fresh and healthier than other available products [81,82]. Notably, we recorded no differences in the socio-demographical characteristics of the respondents and the intrinsic determinants of organic food purchase. 

Moreover, research has also confirmed that the more educated consumers are, the more likely they are to purchase organic products and value sustainability and the environment [48,52,53,54,55], findings backed by the current research. Nevertheless, the price of organic products still remains the main barrier for many consumers and potential consumers, as research suggests [45,46]. Compared with existing data, the results revealed that the higher the income and education level of the respondents, the less likely they were to rely on the extrinsic characteristics of the products. Studies also suggest that increased knowledge and a more affordable price are elements that would, in turn, lead to even more significant growth in popularity and purchase of organic products [69].

The study also aimed to assess the influence of the COVID-19 pandemic on the consumption frequency of organic products. To this extent, the findings are similar to those obtained by other studies in other countries [68,69]. More specifically, the majority of the respondents reported no significant changes in the consumption frequency of organic products (60.5%). It is worth mentioning that 18.4% of the respondents reported lower consumption of organic food, which could be explained by financial insecurities due to the pandemic, food markets being closed, as well as lockdowns which further reduced the mobility and sources of purchasing these types of products.

Nevertheless, respondents who reported frequent consumption of organic products have either increased their consumption frequency or maintained it at the same level, possibly due to clearly established habits. With consumers belonging to the “Once a week” category, the proportions were rather similar regarding the changes in frequency during the pandemic, and responders who consumed organic products more seldom had preponderantly decreased their consumption of these products.

In a more detailed analysis regarding the consumption frequency of various types of organic products after the pandemic, our study has confirmed previous findings, which suggested that the consumption of organic fresh fruits and vegetables has likely remained the same or increased rather than decreased [69]. Moreover, other studies have reported that consumers were more likely to decrease than increase the frequency of meat purchasing, similar to our findings [55,68]. Although at the beginning of the pandemic, the purchase of canned goods has generally increased due to a tendency towards stockpiling [83], our findings reported that, regarding the consumption frequency of organic fruits and vegetables, it has rather remained the same or was more likely to decrease after the pandemic.

When it comes to the limitations of the study, it is worth mentioning that the consumption of organic products, in general, was analyzed rather than focusing on a particular category. Additionally, the consumption behaviors were measured using a questionnaire rather than direct measurements. The sample size does not allow making assumptions about the Romanian consumer in general and how the pandemic has influenced organic food consumption in the whole country. Due to the relatively low number of organic consumers in Romania, compared with other European countries, restrictions of the study could be represented by way of applying the questionnaire and misrepresentation of the targeted consumers. Nevertheless, the results of this pilot study indicate the need for further research into the field of organic products, both in the Northwestern region of Romania as well as countrywide.

## 6. Conclusions

Significant research regarding organic food consumption has identified the main factors influencing consumer behavior when it comes to food purchases. Superior quality, healthy, fresh, without GMOs, and environmentally friendly are some of the most important qualities of organic factors being ranked as the most important ones when purchasing these types of products. 

Thus, the study revealed that consumers tend to pay attention to roughly the same intrinsic qualities of organic products, regardless of their socio-demographical background. Moreover, factors such as education level and income influence the process of decision-making when it comes to extrinsic characteristics of the products, such as price, brand, and labels.

The COVID-19 pandemic has increased health, environmental, and sustainability concerns, with more and more people paying attention when it comes to the food they purchase. To this extent, in light of changing market behavior, promoting the consumption of organic products is also of great importance. With many people having internet access and a social media presence, this could be done by online sales through e-commerce platforms, as well as online shopping, together with Facebook pages and Youtube channels interacting with many potential customers, allowing placing orders with pick-up at fixed points. Additionally, the pack-your-own type of direct sale can further create a long-lasting collaboration between producers and customers, leading to sustainable and responsible consumption. 

With organic products becoming more and more popular, these behavioral changes are likely here to stay, prompting the producers to adapt to a rapidly growing market, all while maintaining the quality of their products. 

## Figures and Tables

**Table 1 ijerph-19-12712-t001:** The socio-demographic profile of the respondents.

Characteristics	Variables	Number of Respondents (N = 190)	% of Respondents
Gender	Female	135	71.1
	Male	55	28.9
Education	High school	48	25.3
	University degree	73	38.4
	Postgraduate degree	69	36.3
Age	18–25 years	36	18.9
	26–35 years	45	23.7
	36–46 years	44	23.2
	46–55 years	45	23.7
	>55 years	20	10.5
Residence	Rural Urban	68 122	35.8 64.2
Monthly income	<2000 RON	14	7.4
	2001–3000 RON	38	20
	3001–4000 RON 4001–5000 RON >5000 RON	40 29 69	21.1 15.2 36.3
Children in household	No Yes	113 77	59.5 40.5

Average exchange rate for August 2022: 4.8953 RON = 1 Euro [77].

**Table 2 ijerph-19-12712-t002:** The two factors resulting from the PCA analysis with the corresponding items.

Eigenvalue	Variance %	Factor	Item	Factor Loading	Mean	SD
11.814	62.180	Intrinsic cues α = 0.895 mean = 4.07 ± 1.098	Fresh	0.895	4.22	1.187
			Healthy	0.890	4.15	1.238
			Natural	0.877	4.07	1.237
			Without additives	0.864	4.02	1.204
			Superior Quality	0.858	4.23	1.211
			Tasty	0.853	4.03	1.232
			Non-Polluting	0.843	4.02	1.188
			Without GMOs	0.832	4.02	1.274
			With a shelf life	0.816	4.17	1.197
			Nutritional value	0.710	3.74	1.245
2.117	11.143	Extrinsic cues α = 0.922 mean = 3.32 ± 1.048	Producer’s logo	0.839	3.18	1.350
			Brand	0.834	3.06	1.302
			Packaging	0.772	2.95	1.344
			Easy to cook	0.757	3.44	1.323
			Variety	0.748	3.52	1.263
			Available in supermarket	0.686	3.51	1.383
			Friends’ recommendation	0.677	3.36	1.268
			Price	0.508	3.65	1.304
			Country of origin	0.462	3.51	1.352
Total variance %	73.323, α = 0.964					

**Table 3 ijerph-19-12712-t003:** Relationship between the socio-demographic characteristics of the respondents and the PCA.

Characteristics Variables		Intrinsic Cues	Extrinsic Cues
Gender	Female	3.99 ± 1.201	3.23 ± 1.043
	Male	4.26 ± 0.765	3.53 ± 1.042
*p*-value		0.815	0.067
Education level	High school	4.02 ± 1.084	3.27 ± 1.084
	University degree	4.12 ± 1.008	3.56 ± 1.043
	Post-University degree	4.04 ± 1.206	3.10 ± 1.020
*p*-value		0.708	0.031 *
Age	18–25 years	4.08 ± 1.012	3.14 ± 0.950
	26–35 years	4.34 ± 0.758	3.55 ± 0.901
	36–45 years	4.01 ± 1.231	3.51 ± 1.123
	46–55 years	3.98 ± 1.145	3.20 ± 1.129
	>55 years	3.75 ± 1.408	2.97 ± 1.070
*p*-value		0.648	0.112
Monthly household income	<2000 RON	3.95 ± 1.104	3.36 ± 1.101
	2001–3000 RON	4.17 ± 1.106	3.75 ± 1.112
	3001–4000 RON	3.94 ± 0.967	3.45 ± 1.014
	4001–5000 RON	3.93 ± 1.331	3.12 ± 0.949
	>5001 RON	4.16 ± 1.072	3.08 ± 1.002
*p*-value		0.342	0.011 *
Children in the house	Yes	4.01 ± 1.114	3.35 ± 1.082
	No	4.10 ± 1.090	3.30 ± 1.030
*p*-value		0.409	0.810
Residency	Rural	3.91 ± 1.157	3.37 ± 1.073
	Urban	4.15 ± 1.201	3.23 ± 1.043
*p*-value		0.101	0.577

* *p* < 0.05

**Table 4 ijerph-19-12712-t004:** Changes in the frequency of consumption after the COVID-19 pandemic.

Changes in Consumption Frequency after the Pandemic	Number of Respondents	% of Respondents
Increased	40	21.1
Decreased	35	18.4
Remained the same	115	60.5

**Table 5 ijerph-19-12712-t005:** Frequency of consuming organic food products before and after the COVID-19 pandemic.

Frequency	Before the Pandemic		Changes in Frequency of Consumption after the Pandemic (%)		
	Number of Respondents	% of Respondents	Increased	Decreased	Remained the Same
Daily	26	13.7	26.9	3.8	69.2
More times a Week	91	47.9	20.9	7.7	71.4
Once a Week	31	16.3	32.3	22.6	45.2
2–3 Times a Month	32	16.8	9.4	46.9	43.8
Once every 6 Months	7	3.7	14.3	57.1	28.6
Less than once every 6 Months	3	1.6	66.7	33.3	0

**Table 6 ijerph-19-12712-t006:** Frequency of consumption of the main categories of organic products investigated after the COVID-19 pandemic.

Type of Product (N = Number of Consumers)	Increased (%)	Remained the Same (%)	Decreased (%)
Dairy (N = 143)	25.2	55.2	19.6
Meat (N = 125)	22.4	53.6	24
Fresh vegetables (N = 175)	20.6	60.6	18.8
Fresh Fruits (N = 179)	20.1	60.3	19.6
Canned Vegetables and Fruits (N = 89)	22.5	46.1	31.4
Cereals (N = 109)	20.1	56	23.9
Eggs (N = 159)	20.7	58.6	20.7
Sweets (N = 83)	19.3	53	27.7

## Data Availability

Not applicable.

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
