# Peer review of "Consumers’ Perception towards Organic Products before and after the COVID-19 Pandemic: A Case Study in Bihor County, Romania"

_ijerph, 2022, doi:10.3390/ijerph191912712_

Round 1
Reviewer 1 Report
1.Originality:
This study has made scientific efforts to evaluate consumers’ perceptions of organic products. However, this article lacks originality in terms of its overall intent and findings, as there have been numerous similar studies. Innovative endeavors should not merely replicate the work of other scholars in a new context but also take into account their theoretical and practical contributions.
2. Relationship to Literature:
The present study reviewed a certain number of prior research, which is commendable. However the authors’ literature review should not be generalized but should have a clear logic and structure, especially the need to provide directional evidence to support the results in later sections. For instance, the authors could have focused primarily on the factors affecting organic food products, the changes in people’s purchasing behavior before and after the COVID-19 pandemic, and their relationships. You’re discussing consumers’ perception towards organic products before/after the covid-19 pandemic, however some updated and related literature were not mentioned/cited yet. I suggest the authors to consider reading and citing all the following updated and impactful literatures (which indicate consumers’ adoption Intention toward new items in different industries/contexts); sure you can find similar impactful & updated papers not limit to listed ones. By doing so you can much enhance hypotheses argument:
· About food consumption during covid-19: Jiao, W., Liu, M., Schulz, P. J., Chang, W. Y, (2022). Impacts of self-efficacy on food and dietary choices during the first COVID-19 lockdown in China, Foods, 11, 2668.
· A large scale investigation (involving 38 countries) about food/meal choices in covid-19: Charlotte De Backer, Lauranna Teunissen, Isabelle Cuykx, Paulien Decorte, Sara Pabian, Sarah Gerritsen, Christophe Matthys, Haleama Al Sabbah, Kathleen Van Royen, Corona Cooking Survey Study Group (2021). An evaluation of the COVID-19 pandemic and perceived social distancing policies in relation to planning, selecting, and preparing healthy meals: an observational study in 38 countries worldwide, Frontiers in Nutrition, 7:621726.
· About consumers’ perception towards green products: Mo. Z., Liu M., Liu, Y. (2018). Effects of functional green advertising on self and others, Psychology & Marketing. 35(5), 368-382.
3. Methodology:
(1) There are also obvious methodological weaknesses in this study. In particular, the sample size of 190 was insufficient to produce convincing results. It is suggested that the authors increase the sample size (at least 300) before reanalyzing the data. In addition, it is worth emphasizing that the small sample size is not a limitation but a flaw. It would be great if you can enlarge the sample size in current study.
(2) The measurements of key variables should be clarified, and information about them should be displayed in detail. In addition, the research design, sampling technique, and data collection procedure must be described.
(3) Table 1 and its accompanying explanation should be relocated to the Results section. Another minor error is, that in line 227, the p-value should be less than 0.001.
4. Results:
(1) The authors categorized 19 factors influencing the consumption of organic food as "intrinsic cues" and "extrinsic cues" based on the results of the PCA. PCA is a data-driven method, so the authors need to back up their categorizations with research literature instead of making up their names.
(2) Since PCA yields two components, confirmatory factor analysis (CFA) should be performed in AMOS to further see if the data can confirm this hypothetical model. All subsequent results are contingent on a satisfactory fit of the model to the data.
(3) Simple frequency counts, such as those in Table 5, cannot determine if changes in the organic food consumption are caused directly by the COVID-19 pandemic. In addition, as previously mentioned, it is unclear how to measure the behavior of consumers towards organic food products before and after the COVID-19 pandemic. This study would have been flawed if it had only asked one question about the level of change (increase, decrease, or the same) without considering how organic foods are categorized.
(4) This study does not validate the relationship between the two component factors and food consumption during the COVID-19 pandemic, which makes this study inadequate in terms of methodological design.
5. Implications for research, practice and/or society:
(1) One separate paragraph needs to be created based on the emphasis on the relevant practical impact on society. According to the results, these implications should be clarified one by one, which may benefit government departments, policymakers, and the public.
(2) The limitation paragraphs must be rewritten and thoroughly considered. As stated previously, the sample size should not be a major concern, as it is in most social science studies, and as we all know, the larger the sample size, the better the results.
6. Quality of Communication:
The manuscript requires moderate English editing. The authors should also clarify why the dataset is unavailable at the end, which would cast doubt on the authenticity of this study. Notably, the authors must explain why the institutional review board and informed consent statements are not applicable, which raises significant ethical issues.
Author Response
Dear Reviewer,
Thank you very much for taking the time and interest into analysing our manuscript. We highly value your thoughtful observations, and we have made our best in order to fulfill these items.
Please see the attachment below for a detailed, point-by-point response.
Best regards,
The authors of the manuscript

Reviewer 2 Report
Dear authors,
I congratulate you on your chosen topic.
My main suggestions are as follows:
I recommend inserting research hypothesis(es) to be validated or invalidated later.
Please measure the sample using an established formula, so that the population of the analyzed area is taken into account and the degree of error is determined.
Please state the limitations of the research.
Please detail and justify the choice of the case study, namely Bihor county.
Author Response
Dear Reviewer,
Thank you very much for taking the time and interest into analysing our manuscript. We highly value your thoughtful observations, and we have made our best in order to fulfill these items.
Please see the attachment below, for a detailed point-by-point response to your observations.
Best regards,
The authors of the manuscript

Round 2
Reviewer 1 Report
I believe the paper can be improved to a published level with further efforts input. Before I recommend ACCEPT, please take my comments as constructive help & follow my suggestions to make the work a better piece to fit the journal’s publication standard. I’m looking forward to seeing an improved manuscript in re-submission round.
Literature review were updated- good to see that. I further suggest the authors to consider adding the following literature to enhance your arguments:
· Page 3: …..Research also revealed that younger consumers tend to be more interested in the environment and the consequences their actions have on it…. maybe cite: Lee, K. (2008), "Opportunities for green marketing: young consumers", Marketing Intelligence & Planning, Vol. 26 No. 6, pp. 573-586
· Page 5: In 3.2, you may further explain why you chose Bihor specifically. I know you mentioned “Bihor county belongs to the Northwestern region of Romania, with an area of 7844 km2, out of which 309.265 hectares of arable land…” but it’s not a sufficient reason. Please further enhance.
· Page 6: I suggest you to offer an example about currency exchange, say 1000 RON= XX USD or HKD so readers can know the info better.
· Page 11-12: You mentioned “The COVID-19 pandemic has increased health, environmental and sustainability concerns, with more and more people paying increased attention when it comes to the food they purchase. With organic products becoming more and more popular, these behavioral changes are likely here to stay, prompting the producers to adapt to a rapidly growing market, all while maintain the quality of their products…” Besides food quality, promotion should be a very important element to encourage people buy more and eat more to enhance their health after pandemic. You may further offer some practical suggestions to promote organic/ethical food/products to consumers, especially to young people or target consumers with specific socio-demographical background through new media approaches (e.g. Wechat, KOL, Vlogger, Youtube).
It’s good to see a separate paragraph about practical impact was added. It’s also good to see improved limitation paragraphs.
It’s good to see the writing of the manuscript was improved. I suggest you to add the following justification either in 3.2. Sample Size, Data Collection, and Organic Food Industry in Bihor County or in Appendix so readers can know there is no problem about data collection and dataset “…Ethical review and approval were waived for this study, due to the fact that participation was voluntary, and all data were anonymous. When it comes to the informed consent, it was obtained from all respondents involved in the study. Using a questionnaire-based method, all respondents had to provide their consent in order to proceed to the actual set of questions”.
Finally, to make sure the publication can be well demonstrated and fast delivered, please hire an English native speaker/proofreader to go through the manuscript again before resubmission.
Author Response
Dear Reviewer,
Thank you very much for your collaboration in improving the quality of our manuscript. Please see the attachment below for a detailed point-by-point response to your observations.
We now hope that our manuscript meets the standards and requirements of this journal.
Best regards,
The authors of this manuscript
